# Investigating the feasibility and acceptability of the HOLOBalance system compared with standard care in older adults at risk for falls: study protocol for an assessor blinded pilot randomised controlled study

Matthew Liston ![ORCID] ,[1] Gregory Genna ![ORCID] ,[2] Christoph Maurer,[3] Dimitris Kikidis,[4] Dimitris Gatsios,[5,6] Dimitris Fotiadis,[5] Doris-Eva Bamiou,[2] Marousa Pavlou[1]

**Correspondence to**
Dr Matthew Liston;
matthew.liston@kcl.ac.uk

## ABSTRACT

**Introduction** Approximately one in three of all older adults fall each year, with wide ranging physical, psychosocial and healthcare-related consequences. Exercise-based interventions are the cornerstone for falls prevention programmes, yet these are not consistently provided, do not routinely address all components of the balance system and are often not well attended. The HOLOBalance system provides an evidence-based balance training programme delivered to patients in their home environment using a novel technological approach including an augmented reality virtual physiotherapist, exergames and a remote monitoring system. The aims of this proof-of-concept study are to (1) determine the safety, acceptability and feasibility of providing HOLOBalance to community dwelling older adults at risk for falls and (2) provide data to support sample size estimates for a future trial.

**Methods** A single (assessor) blinded pilot randomised controlled proof of concept study. 120 participants will be randomised to receive an 8-week home exercise programme consisting of either: (1) HOLOBalance or (2) The OTAGO Home Exercise Programme. Participants will be required to complete their exercise programme independently under the supervision of a physiotherapist. Participants will have weekly telephone contact with their physiotherapist, and will receive home visits at weeks 0, 3 and 6. Outcome measures of safety, acceptability and feasibility, clinical measures of balance function, disability, balance confidence and cognitive function will be assessed before and immediately after the 8 week intervention. Acceptability and feasibility will be explored using descriptive statistics, and trends for effectiveness will be explored using general linear model analysis of variance.

**Ethics and dissemination** This study has received institutional ethical approvals in Germany (reference: 265/19), Greece (reference: 9769/24-6-2019) and the UK (reference: 19/LO/1908). Findings from this study will be submitted for peer-reviewed publications.

**Trial registration number** NCT04053829.

### Strengths and limitations of this study

► This study uses a pilot randomised control design to explore the safety, acceptability and feasibility of providing the HOLOBalance intervention compared with an established home exercise programme.

► The HOLOBalance programme provides an individualised training programme prescribed by a treating clinician which addresses all facets of the postural control system.

► The HOLOBalance study will use quantitative and qualitative data to explore the acceptability of providing holographic balance training programmes to older adults at risk for falls.

► This study will provide crucial insight into the feasibility of implementing state-of-the-art technologies in the home environment.

► This pilot study is not powered to determine whether HOLOBalance reduces falls risk or falls rate.

**Protocol version** V.2, 20 January 2020

## INTRODUCTION

One-in-three people over the age of 65 fall annually.[1] It is widely accepted that falls are multi-factorial in nature[2,3] and that impaired balance function is associated with falls.[4] Maintaining balance is complex and depends on sensory inputs from the visual, proprioceptive and vestibular systems to provide information on body position and motion. Alongside this, central processes monitor and control the interaction between musculoskeletal and neural systems to generate anticipatory postural adjustments and adapt posture to changing environmental and balance task demands.[5] Age-related declines in these systems are well documented and lead to

impaired postural and gait control thus increasing risk for falls.[6 7] Recent studies have identified that vestibular dysfunction is common in older adults who fall,[6 8–13] with up to 80% of fallers experiencing vestibular dysfunction.[9] Although sensory inputs are crucial for postural control, cognitive functions are equally as important for maintaining safe community ambulation. Deficits in executive function and attention are common in older age and are associated with impairments in postural control, dual-tasking balance ability (ie, walk and talk at the same time), reduced gait speed and increased falls risk.[14–20] Additionally, changes in cognitive function strongly predict impairment in activities of daily living and functional independence.[19 21–23] Thus, falls must be considered as multifactorial in nature and should be targeted in interventions that address domains that impact on balance and quality of life.

As falls have wide-ranging physical and psychological consequences and increase the likelihood of frailty, cognitive decline, sedentary behaviour, social exclusion, injury and death,[5–7 21 24–26] their prevention is of primary concern to many public health bodies.[27–29] The National Institute for Health and Care Excellence UK guidelines[30] recommend that older adults at risk for falls should undergo a balance assessment and receive targeted interventions to improve balance and reduce risk for falls. Despite the strength of available evidence, compliance and implementation for providing balance rehabilitation has been poor in UK healthcare settings.[31]

Balance rehabilitation typically consists of personalised sets of exercises, defined by a healthcare professional and is the only effective treatment for balance disorders, irrespective of a person's age.[32–34] Although exercises are typically brief and easy to perform, there is up to 50% lost to follow-up rate in older adults in these programmes, and while supervision significantly increases compliance and effectiveness,[35] it is costly to provide. To this end, many falls prevention programmes provide exercise programmes completed either in group settings or independently at home.[36–41] These programmes have been shown to be effective in reducing falls rates in older adults, reducing falls by 30%–40% (dependant on target population), however, they are limited in that they either do not provide exercises to address vestibular function or dual task training,[37–39] or these are not introduced until very late in the programme.[40 41] This is despite promising evidence suggesting that combined cognitive and functional training may provide improvements beyond single task training.[42–44] Similarly, rehabilitation programmes which address vestibular dysfunction (ie, multisensory rehabilitation, MSR programmes) have been developed for older adults at risk for falls and have shown substantial additional reduction of falls risk to standard programmes.[45–47] These programmes require customised, expert-led interventions to optimise recovery, however, there is limited availability of experts to provide these individualised interventions and these may hinder their translation into clinical practice.

The HOLOBalance telerehabilitation platform has been developed to address this lack of expert physiotherapists. HOLOBalance will provide a customised and interactive falls rehabilitation programme that incorporates: (1) functional balance training, (2) multisensory exercises to improve balance function and (3) cognitive-motor training, for older adults at risk for falls. HOLOBalance will use off-the-shelf technologies to allow clinicians to provide an individualised balance rehabilitation programme for older adults who will be able to participate in at a time of their choice and in their home environment (for system description please visit www.holobalance.eu). This study protocol outlines the multisite randomised controlled proof-of-concept study to explore safety, feasibility and acceptability of the HOLOBalance telerehabilitation system.

## METHODS AND ANALYSIS
### Trial design
This study is an assessor-blinded, randomised proof-of-concept study to explore the acceptability and feasibility of providing a home-based balance telerehabilitation programme (HOLOBalance) to community-dwelling older adults at risk for falls. All participants will be enrolled in a 8-week home exercise programme (HEP) designed to improve balance function, and which adheres to the current guidance for strength and balance training.[30] All participants will receive an exercise programme provided under the supervision of a physiotherapist with regular reviews by telephone (weekly) and home visits (week 0, 3 and 6). Exercises will be prescribed according to participant feedback and task performance as assessed by the treating physiotherapist.

This study will compare acceptability of the HOLOBalance telehealth programme (eg, compliance, drop-out rate) to an established HEP that is routinely used in community rehabilitation (the OTAGO HEP[23]) and will explore trends for effectiveness across a number of validated outcome measures to explore whether a future trial is warranted, and if so to provide data for a sample size estimate. For this proof-of-concept study, data will be collected at baseline (week 0) and at completion of the intervention (week 9). The flow of participants through the trial will be recorded in compliance with the Consolidated Standards of Reporting Trials statement (figure 1), and this protocol has been developed using the Standard Protocol Items: Recommendations for Interventional Trials reporting guidelines for clinical trials.[48]

### Participants
One hundred and twenty independently living, community-dwelling older adults aged 65–80 who are at risk of falls will be recruited across three sites (London (UK), Freiberg (Germany), Athens (Greece)). Older adults who meet the following inclusion criteria for entry will be eligible to be enrolled.

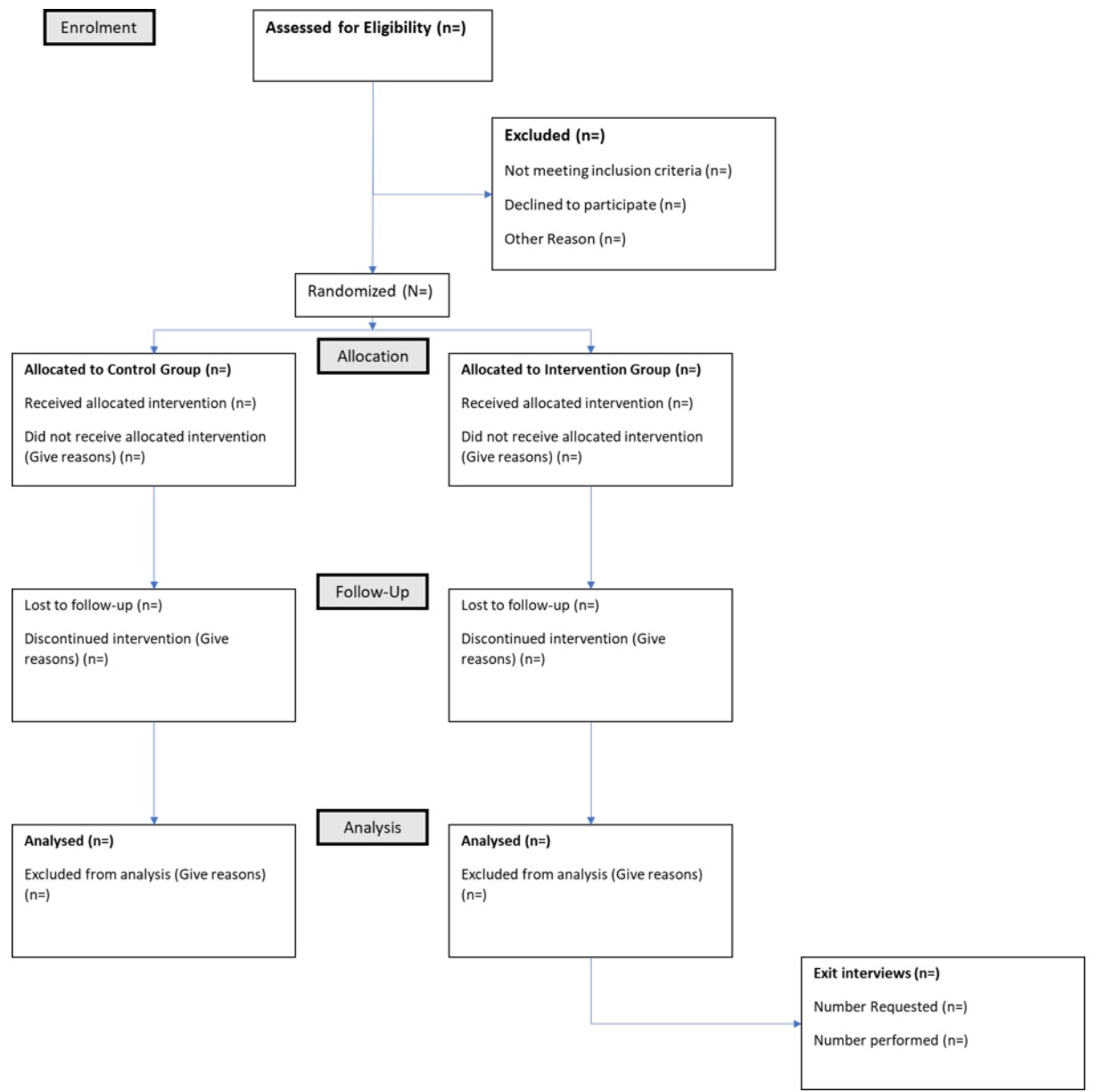

**Figure 1** CONSORT diagram to demonstrate participant flow through the HOLOBalance study. CONSORT, Consolidated Standards of Reporting Trials.

## Inclusion criteria
► Independent community-dwelling participants able to walk 500 m independently or with a stick.
► No significant visual impairment.
► Able to understand and to consent to the research.
► A score of >22 on the Montreal Cognitive Assessment (MoCA), that is, adults with no or mild cognitive impairment.
► At risk of falls (ie, Functional Gait Assessment, FGA less than 22/30), and / or have significant fear of falling (Falls Efficacy Scale International (FESI) short form >10) and /or have experienced a fall/s in the last 12 months.
► Willing to participate and to comply with the proposed training and testing regimen.

► Available space of 1×2 m at home and sufficient home broadband to allow the system to operate as designed.

## Exclusion criteria
► Orthostatic hypotension or uncontrolled hypertension.
► Have depression that is, a score of >10 at the Geriatric Depression Scale.
► Have cognitive impairment as indicated by the MoCA score(score <22).
► Other neurological problem (eg, stroke, Parkinson's disease, peripheral neuropathy).
► Acute musculoskeletal injury that prevents participation in a structured exercise programme (eg, lower limb fracture).
► No internet connection at home.

- ► Has participated in a clinical drug trial in the past 6 months.
- ► Currently receiving falls and/or cognitive rehabilitation.
- ► Has an implanted medical device or cardiac pacemaker.

## Recruitment

Community-dwelling older adults will be recruited via adverts placed in the community, on social media and by referrals made by recruiting clinics. Potential participants will undergo a telephone screening session with a researcher to determine their eligibility, with eligible participants booked in for baseline testing with the blinded outcome assessor. Participants will provide written informed consent on arrival at their baseline outcome assessment session.

## Randomisation

Patients will be randomised using an online platform (www.sealedenvelope.com) into either the intervention or control groups. Participants will be allocated in blocks of 10 with an allocation ratio of 1:1 to control (OTAGO HEP) and intervention (HOLOBalance) groups. Randomisation procedures will be run independently for each site and will be completed by a researcher external to the research study. No out of hours randomisation is required for this study.

Allocation will be concealed in consecutively numbered opaque envelopes which will be drawn up for each block and presented to the treating physiotherapist after the baseline assessment session has been completed. Allocation will not be revealed until after the baseline assessment has been completed, and once revealed will be entered onto the enrolment log.

## Blinding

This is a single (assessor) blinded proof-of-concept study. As this study is comparing a technology intervention (HOLOBalance) versus standard care it is not possible to blind the participants or the treating clinicians to the intervention provided. The blinded outcome assessor will collect all of the outcome measures at baseline and follow-up with a single assessor being used at each site. Each assessor will be required to attend a training session prior to data collection to ensure consistency across sites. The outcome assessor will be asked to record any incidences of unblinding and detail how this occurred. Due to the nature of this study, we do not foresee circumstances where emergency unblinding of the outcome assessor is required.

## Interventions

Participants will be randomised to receive either the active intervention (HOLOBalance) or the control intervention (OTAGO HEP). Participants will be asked to perform their prescribed exercises daily, with each participant performing 40–60 min of rehabilitation per day (including rest breaks). Participants will receive home

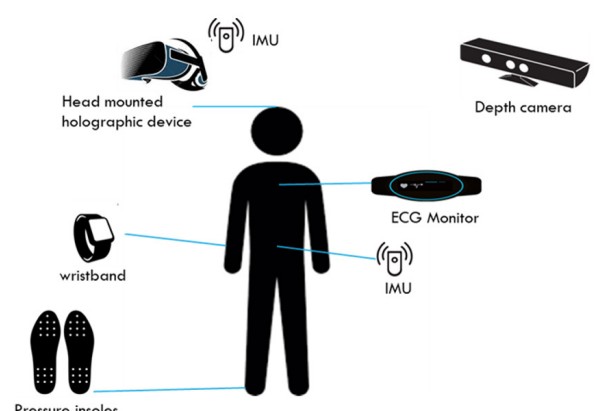

**Figure 2** Diagram of the HOLOBalance system hardware.

visits from their treating clinician at week 0, 3 and 6 to assess and progress their programme. All participants will have a telephone conversation with their treating clinician every week for the duration of their participation in the rehabilitation programme to discuss their exercise programme.

## Active intervention: the HOLOBalance system

HOLOBalance is an augmented reality balance training programme based on evidence-based MSR protocols.[49 50] MSR requires individuals to regularly perform exercises which challenge the balance system (eg, closed eyes, stand on foam), optimise vestibular balance function and have been shown to improve balance control in healthy older adults[49 50] and people with vestibular balance dysfunction.[51–54]

Participants will be required to wear a series of body worn sensors (pressure detecting insoles, intertial measurement unit (IMU) and a heart rate monitor, figure 2) when performing the exercises so that (1) the computer software can assess the participants performance and (2) the user can interact with the system. HOLOBalance will provide prescribed exercises and exergames presented into the person's home environment by ahead mounted augmented reality display and log the user's interactions with the system (eg, number of sessions completed). HOLOBalance will display a holographic virtual physiotherapist (figure 3A) to provide instructions and demonstrate prescribed exercises. The HOLOBalance programme will include exercises to maximise opportunities for balance improvement and include turning the head while looking at a target, standing on foam, and walking while turning the head (eg, of gameplay, please see figure 3B,C, and for table of basic exercises please see table 1). In addition, participants will be given auditory and cognitive exercises in supplement to their balance training programme. For further information on the HOLOBalance intervention, please visit www.holobalance.eu.

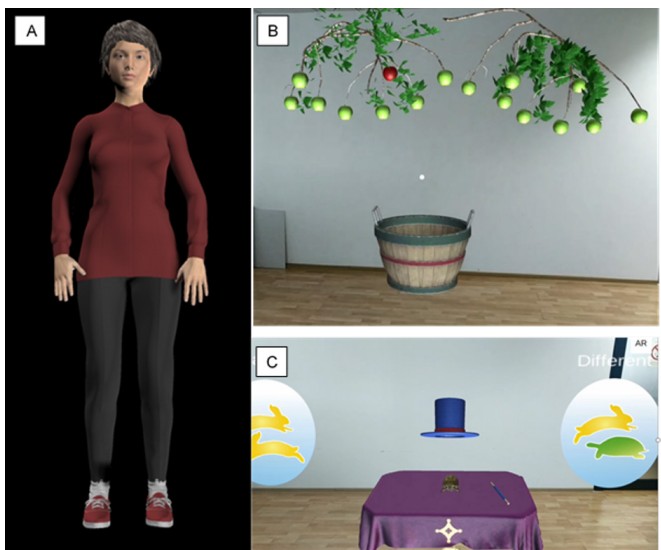

**Figure 3** Screenshots of (A) physiotherapist avatar, (B, C) representative gameplay from the HOLOBalance system.

## Control intervention: the OTAGO HEP

The OTAGO HEP booklet (available at https://www.livestronger.org.nz/assets/Uploads/acc1162-otago-exercise-manual.pdf) will be provided to all individuals in the control arm. Following the physiotherapy home assessment, participants will be provided with a list of exercises from the OTAGO booklet for them to complete every day. Participants will be required to log their exercises in an exercise diary for review by the treating clinician at the next home visit. Exercises will be progressed in line with the recommendations of the OTAGO HEP in combination with the treating clinician's clinical judgement.

## Outcome measures

### Primary outcomes

The primary outcomes will relate to the acceptability, safety and feasibility of providing the HOLOBalance intervention. Acceptability measures include; recruitment rate (percentage of eligible participants enrolled), compliance with interventions (percentage of prescribed sessions completed in HOLOBalance and control groups) and drop-out rates (number of drop-out in HOLOBalance and control groups). Exit interviews will be completed with participants that receive the HOLOBalance intervention to explore their experience of using the system, including perceived benefits, frustrations and recommendations. Interviews will be recorded, transcribed verbatim and subject to thematic analysis.

In the HOLOBalance group, adherence will automatically be monitored through the interaction with the system, and calculated as the number of prescribed balance exercises, auditory and cognitive training tasks completed per day, expressed as a percentage. In the control group, participants will be required to complete a daily exercise diary to report on their level of activity, whether they had completed their HEP session and whether they had performed their cognitive tasks. All

participants will receive a weekly telephone call from the research physiotherapist to discuss any issues, their current progress and to remind them to complete their exercise diaries (if in the control group) which will be collected at their final outcome measure test session. The order of events for participants enrolled in this study is displayed in table 2.

Safety of the HOLOBalance intervention will be assessed by recording and assessing all adverse/serious adverse events and device effects (and reporting these to appropriate agencies as required). These will be reported by participants to their treating clinician or by self-reporting on the HOLOBalance database. The feasibility of the study will be explored by collecting information on and analysing any deviations from protocol or problems with implementing the study protocol (eg, logistical problems).

## Secondary outcomes

### Balance assessment

The mini balance evaluation systems test[55] (mini BESTest) is a 14-item test that assesses dynamic balance components including anticipatory postural adjustments, reactive postural control, sensory orientation and dynamic gait. The data are based on a total score of 28 points. The test takes approximately 10 min to complete. This will be collected at baseline (week 0) and follow-up (week 9)

The Functional Gait Assessment[4] (FGA) is a 10-item test that assesses performance on complex gait tasks (eg, walking with head turns or stopping and turning). The test takes approximately 5 min to complete. Scores of 21 or less indicate risk for falls.[4] This will be collected at baseline (week 0) and follow-up (week 9).

Falls diaries will be used to record whether participants have fallen each day, with falls defined as a person coming to rest inadvertently on the ground or floor or other lower level. Participants will return falls diaries monthly for the duration of the intervention and for 6 months after completion.

### Cognitive assessment

The validated Montreal Cognitive Assessment[56] (MoCA): This test includes sections on visuospatial/executive function, naming, attention, language, abstraction and orientation to time and place. This test will be used as a screening test for cognitive function prior to inclusion in the trial. This will be collected at baseline (week 0) and follow-up (week 9).

The validated Cambridge Neuropsychological Test Automated Battery[57] (CANTAB) cognitive test battery will be used to assess specific cognitive function domains pre-post intervention. The test battery includes: (1) Motor screening task, (2) Paired Associated Learning, (3) Spatial Working Memory, (4) Reaction Time, (5) Rapid Visual Information Processing and (6) Delayed Matching to Sample. This assessment will take approximately 35 min to complete. Processing speed and

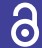

**Table 1** The HOLOBalance intervention basic exercises and their progressions

| Position | Description | Progressions | Progression activation |
|---|---|---|---|
| Seated | Head turns side to side (yaw rotation of 30°) while visually fixating on a static target placed at eye level in front of the participant | Increase speed of movements Visual target moves in opposite direction to head movement | Increase speed of head turns if patient reporting no symptoms If patient can perform three head turns (ie, left to right) in one second without provoking symptoms, move to next progression |
| Seated | Had turns up and down (pitch rotation of 30°) while visually fixating on a static target placed at eye level in front of the participant | Increase speed of movements Visual target moves in opposite direction to head movement Look up to ceiling and down to floor without fixation | Increase speed of head turns if patient reporting no symptoms If patient can perform three head turns (ie, left to right) in one second without provoking symptoms, move to next progression |
| Seated | Bend down to pick object up from the floor in front of you | Close the eyes Reach to pick up objects to the side | If patient can complete the full range of movement consistently in 2 s (or less) in eyes open condition, and no symptoms are provoked. Progress to eyes closed. When able to complete forward bends with eyes open and closed, progress to sideways bend. |
| Standing | Stand with feet hip width apart, looking straight ahead with eyes open | Bring feet closer together Close the eyes | If patient can maintain stability in the correct position for 1 min with eyes open without prompts, then ask patient to close eyes. If patient can maintain stability in the correct position with eyes closed, reduce base of support. |
| Standing | Stand on foam cushion with feet hip width apart and looking straight ahead with eyes open | Bring feet closer together Close the eyes | If patient can maintain stability in the correct position for 1 min with eyes open without prompts, then ask patient to close eyes. If patient can maintain stability in the correct position with eyes closed, reduce base of support. |
| Standing | Stand with feet hip width apart and bend over to pick up an object from the floor | Bring feet closer together Reach up as if to reach into a cupboard | When patient can safely complete full movement in less than 3 s, and this produces no symptoms, ask patient to increase speed. When patient can safely complete the full movement (without symptoms) in under 2 s, move on to the next exercise. |
| Standing | Step turn through 180° to face the opposite direction | Increase speed of movement | Once patient is able to perform the exercise safely and confidently at their baseline speed, ask them to increase their speed. |
| Walking | Walk forwards looking straight ahead | Increase walking speed | Once patient is able to perform the exercise safely and confidently at their baseline speed with no deviation, ask them to increase their gait speed. |

Continued

| Table 1 | Continued | | |
|---------|-----------|---|---|
| Position | Description | Progressions | Progression activation |
| Walking | Walk forwards across the room looking between two targets placed at eye level, 1.5 m apart on the horizontal plane. | Increase walking speed Increase head turn speed Turn head fully to side as if crossing the street. | Once patient is able to perform the exercise safely and confidently at their baseline speed with no symptoms or deviation, ask them to increase their gait speed. Once patient is able to perform the exercise safely and confidently with no symptoms or deviation, ask them to increase their head turn speed Once patient is able to perform the exercise safely and confidently with no symptoms or deviation, ask them to increase amplitude of head turns |
| Walking | Walk across the room and nod your head to look up to the ceiling and down to the ground. | Increase walking speed Integrate diagonal head movements | Once patient is able to perform the exercise safely and confidently at their baseline speed with no symptoms or deviation, ask them to increase their gait speed. Once patient is able to perform the exercise safely and confidently with no symptoms or deviation, ask them to increase their head turn speed Once patient is able to perform the exercise safely and confidently with no symptoms or deviation, ask them to integrate diagonal head movements |

executive functions from the CANTAB suite (http://www.cambridgecognition.com/academic/cantabsuite/battery) will be compiled to derive an executive function score. This will be collected at baseline (week 0) and follow-up (week 9)

### Physical Activity and Social Participation Assessment

The Rapid Assessment of Physical Activity[58] (RAPA) is a 9-item, self-administered questionnaire developed to provide an easily administered and interpreted means of assessing levels of physical activity among adults older than 50 years. RAPA evaluates a wide range of physical activity level, from sedentary to vigorous activity, as well as strength and flexibility training and takes 5 min to complete. It has adequate convergent validity and good criterion validity. This will be collected at baseline (week 0) and follow-up (week 8)

The WHO Disability Assessment Schedule V.2.0[59] is an assessment which provides a global measure of disability. It covers the following domains of functioning: Cognition—understanding and communicating; Mobility—moving and getting around; Self-car—hygiene, dressing, eating and staying alone; Getting along—interacting with other people; Life activities—domestic responsibilities, leisure, work and school; Participation— joining in community activities. This will be collected at baseline (week 0) and follow-up (week 9)

### Subjective questionnaires

The Activities-specific Balance Confidence Scale[60] assesses patient's perceived confidence for performing 16-activities of daily living without losing balance. Scores ≤67/100% indicate increased falls risk. This will be collected at baseline (week 0) and follow-up (week 9)

The Falls Efficacy Scale - Internation (FES-I) Short Form is a short, easy to administer tool measuring an individual's level of concern regarding falling during social and physical activities inside and outside the home, whether or not the person actually does the activity. Level of concern is measured on a 4-point Likert scale (1=not at all to 4=very). It has excellent internal validity and test–retest reliability. Scores >10 for the short form have been suggested as cut points for indicating high concern about falling.[61] This will be collected at baseline (week 0) and follow-up (week 9)

The EQ-5D[62] is a standardised, valid and reliable simple, generic measure of health status for clinical and economic appraisal. The EQ-5D-5L has five dimensions (mobility, self-care, usual activities, pain/discomfort, anxiety/depression) and includes the EQ Visual Analogue Scale (EQ VAS). The respondent is asked to rate their health status on these five dimensions from 1 to 5, respectively, as no problems, slight problems, moderate problems, severe problems and extreme problems. The EQ VAS records the respondent's self-rated health on a 20 cm vertical, VAS



**Table 2** Order of events for all participants enrolled in the HOLOBalance study

| | | Screening session | Baseline test session | Week 0 | Daily | Weekly | Monthly | Week 3 | Week 6 | Follow-up test session | Week 9 |
|---|---|---|---|---|---|---|---|---|---|---|---|
| Recruitment Procedures | Provide patient information sheet | X | | | | | | | | | |
| | Provide informed consent | | X | | | | | | | | |
| University-based data collection | Demographics | | X | | | | | | | | |
| | The MoCA | | X | | | | | | | X | |
| | The RAPA | | X | | | | | | | X | |
| | The WHODAS 2.0 | | X | | | | | | | X | |
| | The CANTAB | | X | | | | | | | X | |
| | The ABC Scale | | X | | | | | | | X | |
| | The FES-I | | X | | | | | | | X | |
| | The EQ-5D | | X | | | | | | | X | |
| | The EMA | | X | | | | | | | X | |
| | The BREQ-3 | | X | | | | | | | X | |
| | Mini-BESTest | | X | | | | | | | X | |
| | The FGA | | X | | | | | | | X | |
| | Exit Interviews (HOLOBalance Group Only) | | | | | | | | | | X |
| Usability Measures | The SUS | | | | | | | | X | | |
| | The UEQ | | | X | | | | X | X | | |
| Home-based interactions | Equipment installation and Removal (HOLOBalance Only) | | | X | | | | | | X | |
| | Home visit from Physiotherapist | | | X | | | | X | X | | |
| | Telephone conversation with physiotherapist | | | | | X | | | | | |
| | Perform Home Exercise Programme | | | | X | | | | | | |
| | Submit falls diaries | | | | | | X | | | | |

ABC, Activities Specific Balance Confidence Scale; BREQ-3, Behavioural Regulation in Exercise Questionnaire; EMA, Environmental Analysis of Mobility Scale; FES-I, Falls Efficacy Scale International; FGA, Functional Gait Assessment; mini-BESTest, Mini Balance Evaluation Systems Test ; MoCA, Montreal Cognitive Assessment; RAPA, Rapid Assessment of Physical Activity; SUS, System Usability Scale; UEQ, User Experience Questionnaire; WHODAS 2.0, WHO Disability Assessment Schedule 2.

with endpoints labelled 'the best health you can imagine' and 'the worst health you can imagine'. The respondent is asked to mark an X on the scale to indicate 'how your health is TODAY'. This will be collected at baseline (week 0) and follow-up (week 9)

Environmental Analysis of Mobility[63] scale is a self-report scale assessing the effect of the physical environment on community mobility. Twenty-four features of the physical environment are identified. For each feature, an encounter question (How often do you?) is paired with an avoidance question (How often do you avoid?). Subjects report on frequency of encounter and avoidance behaviour using a 5-point ordinal scale (never, rarely, sometimes, often, and always). The test–retest reliability of the questionnaire is good. This will be collected at baseline (week 0) and follow-up (week 9)

The Behavioural Regulation in Exercise Questionnaire[64 65] is a 24-item questionnaire to assess motivation to exercise. Participants rate whether statements apply to themselves (or not) using a 5-point Likert scale ranging from 0 (not true for me) to 4 (very true for me). This will be collected at baseline (week 0) and follow-up (week 9)

### User experience evaluation (HOLOBalance group only)

The System Usability Scale[66] is a 10-item questionnaire that asks individuals to rate their experience of using the system on a 5-point Likert scale ranging from 1 (strongly disagree) to 5 (strongly agree). Example items include 'I think that I wold like to use this product regularly'. This will be collected by the treating physiotherapist at week 6 only

The User Experience Questionnaire[67] is a 26-item self-report questionnaire to assess a person's experience of using a product. Participants rate the product between two contrasting attributes (eg, attractive vs unattractive) on a 7-point Likert scale. This will be collected by the treating physiotherapist at weeks 1, 3 and 6.

The NASA Task Load Index[68] is a widely used, subjective, multidimensional assessment tool that rates perceived workload in order to assess a task, system or team's effectiveness or other aspects of performance. The NASA TLX assesses work load on five 21-point VASs ranging from 0 (very low) to 21 (very high) for task demands including mental demand, physical demand and temporal demand. Questions from the NASA TLX will be integrated into the HOLOBalance tasks. These questions will be administered 20 times per participant in total

### Data management and statistical analysis

Anonymised data will be entered into a study database. The integrity of this data will be assessed by taking a random sample (10%) from the study database (after the completion of data collection) and comparing with the paper case report form (CRF). If there is less than 98% agreement between the paper CRF record and inputted measures, all data will be checked and rectified. The rectified database will be saved under a new filename (eg, studydatabase_rectified_date) and all changes made to the database will be logged. Essential trial documentation will be kept with the Trial Master File and Investigator Site Files. The sponsor will ensure that study documentation is retained in accordance with their local approvals after the conclusion of the trial.

As this is a proof-of-concept study, there is no formal plan for statistical analysis due to the small sample size and lack of statistical power. However, for a future trial an analysis plan has been developed which will be run in this study to determine its appropriateness. We will use intention to treat analysis with repeated measures general linear model analysis of variance for selected outcome measures and other appropriate tests for between-group and within-group analysis. We will also include correlations analysis and $\chi^2$ assumption testing.

### Sample size

This proof-of-concept study will recruit 60 participants per group (ie, total sample size of 120). This will allow the researchers to gain preliminary data on the primary outcomes of safety, feasibility and acceptability of the HOLOBalance intervention. This will also provide sufficient data to explore trends for effectiveness and to allow for sample size estimates to be drawn up for a future trial.

### Study oversight

Trial steering committees and trial management groups will be formed to oversee the conduct of the research according to predetermined terms of reference.

### Patient participant involvement

As part of the larger HOLOBalance project, the research consortium has performed extensive patient participant involvement with older adults, discussing the study design, exergames and the supporting technologies with approximately 75 older adults in total.

## ETHICS AND DISSEMINATION

This study has received institutional ethical approvals in Germany (reference: 265/19) and Greece (reference: 9769/24-6-2019) and from the Health Research Authority in the UK (reference: 19/LO/1908). All amendments to the study protocol which may impact on the integrity of the study, or the data are required to receive approval by the research ethics committee prior to their implementation.

Findings from this study will be provided as lay reports to participants, disseminated to community groups via community partners and will be submitted for peer-reviewed publications. Additionally, electronic data will be anonymised and uploaded to a data repository (Zenodo) that supports restricted access. Electronic data will not be made publicly available and access will to the dataset will only be provided by the data management board of the HOLOBalance research consortium. Use and reuse of the pilot dataset will be subject to the licence under which the data objects were deposited.

**Author affiliations**
¹Centre for Human and Applied Physiological Sciences, King's College London, London, UK
²The Ear Institute, University College London, London, UK
³Department of Neurology and Neuroscience, Medical Center, University of Freiburg, Freiburg im Breisgau, Germany
⁴National and Kapodistrian University of Athens, Athens, Greece
⁵Unit of Medical Technology and Intelligent Information Systems, Department of Material Science and Engineering, University of Ioannina, Ioannina, Greece
⁶Department of Neurology, Medical School, University of Ioannina, Ioannina, Greece

**Contributors** ML, GG, D-EB and MP were involved in the conception, design and writing and editing of the study protocol. DG, DF, DK and CM were involved in the conception and editing of the protocol. All authors approved the final protocol.

**Funding** This project has received funding from the European Union's Horizon 2020 research and innovation programme under grant agreement no 769574.

**Disclaimer** This article reflects only the author's view and the Commission is not responsible for any use that may be made of the information it contains.

**Competing interests** None declared.

**Patient and public involvement** Patients and/or the public were involved in the design, or conduct, or reporting, or dissemination plans of this research. Refer to the Methods section for further details.

**Patient consent for publication** Not required.

**Provenance and peer review** Not commissioned; externally peer reviewed.



**ORCID iDs**
Matthew Liston http://orcid.org/0000-0002-9694-6268
Gregory Genna http://orcid.org/0000-0001-6564-3101

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
