## [Reviewer comments · BMJ Open]

ARTICLE DETAILS

TITLE (PROVISIONAL)	Investigating the Feasibility and Acceptability of the HOLOBalance System Compared to Standard Care in Older Adults at Risk for Falls: Study protocol for an assessor blinded pilot randomised-controlled study.
AUTHORS	Liston, Matthew; Genna, Gregory; Maurer, Christoph; Kikidis, Dimitris; Gatsios, Dimitris; Fotiadis, Dimitris; Bamiou, Doris-Eva; Pavlou, Marousa

VERSION 1 – REVIEW

REVIEWER	Monica Perracini City of Sao Paulo University
REVIEW RETURNED	27-Apr-2020

GENERAL COMMENTS	This is a very relevant and promising intervention conducted within a research network. Data from each of the technologies involved might offer valuable information on tailoring (progress) balance exercises for optimizing participants skills and outcomes. Furthermore, it can also help to understand how to motivate and better engage participants to long-term adherence to exercise to prevent falls. There are some further improvements in order to make the manuscript more robust. Page 4 - Line 9: Recent Cochrane Review (Sherrington et al. Exercise for preventing falls in older people living in the community. Cochrane Database of Systematic Reviews 2019, Issue 1) recommended that multiple exercises (that include resistance exercises) reduce the rate of fall and decrease the rate of falls and the number of people sustaining a fall with a moderate-certainty evidence. The exercises that have high-certainty evidence are balance and functional ones. Suggest reviewing the state that balance and strength is the cornerstone of fall prevention. Page 4 - Lines 20-21: review the sentence construction Page 6 - Line 7: In fact falls are multifactorial in nature. Not only balance, but also gait problems increase the risk of falling. I suggest provide adequate reference, since refs 1 and 2 are not from epidemiological risk factors studies. Page 6 - Line 15: Gait control or walking ability? Ambulatory control seems not appropriate. Page 6 - Line 17: Recommend revising the terminology of inner ear balance dysfunction/exercise (all over the manuscript), since
---

	inner ear is an anatomical site, and the vestibular system is much more complex. It includes peripheral and central structures and functions. Particularly the compensation of vestibular dysfunctions is primarily dependent upon central processes. Page 6 - Line 18 - 19: The value of 80% might be inflated. Although the article by Liston et al., 2014 is a well-designed pilot study, the sample size is very small. Page 6 - Line 48: Balance Physiotherapy is not a commonly used term (?). Consider revising to physiotherapy intervention targeting balance dysfunction or structured balance exercise program. Page 6 - Line 51: Overall, balance training may be easily performed if the applied protocol is simple, such as the Otago program. The prescription and execution of progressive balance exercises (intensity) is challenging, and include a combination of multisensory inputs, neuromuscular and coordination exercises, anticipatory postural adjustments exercises, and dual task tasks. Balance reactive strategies can be trained only in face-to-face interventions. Commonly, no all balance dimensions are accomplished in balance programs to prevent falls. Ultimately, I would recommend the authors to be cautious with the statement that balance exercises can be easily delivered or performed. Page 7 - Lines 3 - 4: Vestibular rehabilitation exercises are targeted to patients with vestibular problems. But, multisensory training that includes exercises that challenge the vestibular system by stimulating the integration of sensory information (re weighting of sensory inputs visual, vestibular and somatosensory) are usually included in balance programs during balance, gait and functional tasks. The problem is that these exercises programs are quite poorly described in the RCTs. Please clarify what is the rationale for including specifically exercises targeting the vestibular rehabilitation for older fallers or clarify if these exercises would be in fact mutisensory training. . Page 7 - Line 19: Clarify what are the health care needs. Otago program is a well-recognized and effective intervention to prevent falls, and for this reason was chosen as a comparator (standard care) in this study. The program is based on five home visits of a physio or nurse and requires a relatively small investment to be implemented. However, overall around 37% of participants remained exercising 3 times per week and 56% 2 times per week, bellow that what is recommended (80%). There are many barriers for attending group-sessions, and the supplementation of group sessions with additional home-based exercise is a strategy used in several effective trials for falls prevention. Alongside, the many technologies involved in the Holobalance platform it seems that one additional advantage would be keeping a long-term adherence. Page 8 - Lines 45 - 46: Please clarify if the 2 situations must be simultaneously present (FGA less than 22/30) and have fear of falling. Provide reference for the cut-off point as being one that reflects risk of falling. Page 8 - Line 51 - Please be more specific by what is meant by no significant visual, musculoskeletal or neurological dysfunctions.
--	---

	What specific conditions and or diseases will be excluded? (Parkinson disease, stroke, neuropathic diabetes) Page 8 - Line 53: Any other acute problem/decompensation (such as acute vertigo, atrial fibrillation, congestive cardiac failure, etc) preventing exercises prescription. As this study is a mutisensory intervention, it would be important to state if this pilot study will include only participants with chronic dizziness (and provide a definition). Page 9 - Line 3: Depression is very common among people at high risk of falling. I agree that the engagement in a digital platform needs drive and motivation, but this is also a limitation that should be further discussed. Page 10 - Line 17: Please provide some information regarding entire duration of the session and particularly how much time will be spent in exercises in the orthostatic position; rest periods; and what kind of safety instructions the participants will receive (for example during the exercise that demands staying in a foam with eyes closed). Authors reported in the introduction that dual task activities would be accomplished in the HOLOBalance program. Please provide at least one or two examples of dual task exercises included. What kind of auditory tasks will be included (rhythmic)? Please provide information regarding how participants will be familiarized with the technologies. It's expected that many of them will not have experienced exergames devices or augmented virtual reality devices. Page 11 - Line 29: Clarify which questions will be addressed in the exit interviews and if some narrative analysis will be conducted with them. Page 11 - Line 60: All secondary outcomes will be used to make between group comparisons? This is a proof of concept study and the effectiveness of HOLOBalance will not be measured. If, a future clinical trial will be conducted would be important to define clinical important differences or meaningful differences for important outcomes rather than effect size. Clinical variables, such as the number of medications, use of psychotropic and medications, numbers of diseases, number of falls in the previous 12 months is going to be reported? Page 15 - Lines 59 - 60: Which measures and cut-off points are going to be used to stratify subgroups. Please clarify what would be the content of the validation report. Authors did not provide a discussion section with the possible impacts and clinical implications of their study. Table 2 would be better displayed in a study flowchart.
--	---

REVIEWER	Eric Anson University of Rochester, NY USA
REVIEW RETURNED	12-May-2020

GENERAL COMMENTS	The manuscript "Investigating the Feasibility and Acceptability of the HOLOBalance System Compared to Standard Care in Older Adults at Risk for Falls: Study protocol for an assessor blinded
---

	pilot randomised-controlled study" is a well written description of a proposed clinical trial investigating the feasibility and acceptability of HOLOBalance for training balance in older adults within their home. This is an important study addressing a need within the aging community. There are a few minor points that will benefit from further clarification which are detailed below: Page 5 lines 7-11 the 2 phrases are poorly connected, please re-write to improve clarity. Page 5 lines 20-26, if possible please re-write into 2 sentences. The length of this statement as written makes it hard to follow. Methods page 6, are the authors going to screen prospective participants with the FGA, or is the FGA routinely administered in the settings from which subjects will be recruited? Methods page 7 line 10-11 - please clarify and consider the potential for sample bias that this specific inclusion criteria creates Methods page 7 lines 42-45, this statement is unclear Methods page 8 lines 3-13, please clarify how many outcome assessors will be used at each site, and what if any training each assessor will have. Also please clarify what steps have been taken to ensure similar grading for instruments that are open to interpretation because of subjectivity. Page 9, lines 49-50 please clarify how and when adverse events will be identified. Methods Page 10, please provide information on psychometrics of the mini-BEST and FGA relevant to the study population if available. Methods Page 10, please clarify the definition of a fall. Methods Page 10, please clarify that validated version of the MoCA exist in the languages appropriate for the proposed study populations. Same comment for CANTAB, and all subsequently listed questionnaire based outcome measures. Methods Page 10, what is "MCI169" Methods page 13, perhaps "entered" instead of "inputted" Methods page 13 lines 52-57, perhaps consider consolidating this statement to something like "Sex/gender will be considered as a covariate biological variable for all analyses. There is a discrepancy in the proposed sample size on page 6 (120) and page 13 (80) and page 14 (120), please clarify. Page 14, PPI and REC are used without being defined. Page 14, please clarify the role of the funding agency/industry partner with respect to data dissemination. Specifically clarify whether the "Holobalance data management board" has any oversight regarding approval of data/results dissemination. Table 1: Seated gaze stabilization exercises, please clarify the target distance for pitch and yaw. Please clarify if the "Step turn through 180°" is a voluntary or cued turn. Table 1, this progression is unclear "Increase head turn amplitude when visual targets are removed," how will the subject know how far to turn if the targets disappear.
--	--

REVIEWER	Yoshiro Okubo Neuroscience Research Australia Australia
REVIEW RETURNED	12-May-2020

GENERAL COMMENTS	This manuscript describes the study protocol for the pilot feasibility trial using the augmented reality program called HOLOBalance involving remote coaching of strength and balance training. The
---

	intervention has great novelty, the manuscript has been written very well and worth publishing in this journal. I have a few minor comments. I believe that reproducibility is an important aspect of a scientific article. Information on the development or reference to it would be beneficial. If the authors do not wish to publish such information due to intellectual property, their plan on how to make the final product accessible can be mentioned. Please include a reference for the OTAGO HEP booklet and the frequency of home visits to the control group participants. Readability of Figure 2 should be improved. The trial was due to start in April 2020, please provide an update on this.
--	--

VERSION 1 – AUTHOR RESPONSE

Reviewer(s)' Comments to Author:

Reviewer: 1

Reviewer Name: Monica Perracini

Institution and Country: City of Sao Paulo University

Please state any competing interests or state 'None declared': None declared

Please leave your comments for the authors below

This is a very relevant and promising intervention conducted within a research network. Data from each of the technologies involved might offer valuable information on tailoring (progress) balance exercises for optimizing participants skills and outcomes. Furthermore, it can also help to understand how to motivate and better engage participants to long-term adherence to exercise to prevent falls. There are some further improvements in order to make the manuscript more robust.

Page 4 - Line 9: Recent Cochrane Review (Sherrington et al. Exercise for preventing falls in older people living in the community. Cochrane Database of Systematic Reviews 2019, Issue 1) recommended that multiple exercises (that include resistance exercises) reduce the rate of fall and decrease the rate of falls and the number of people sustaining a fall with a moderate-certainty evidence. The exercises that have high-certainty evidence are balance and functional ones. Suggest reviewing the state that balance and strength is the cornerstone of fall prevention.

This section has been changed and now reads: "Exercise-based interventions are the cornerstone"

Page 4 - Lines 20-21: review the sentence construction

This section has been altered to improve clarity.

Page 6 - Line 7: In fact falls are multifactorial in nature. Not only balance, but also gait problems increase the risk of falling. I suggest provide adequate reference, since refs 1 and 2 are not from epidemiological risk factors studies.

This has been altered to

One in three people over the age of 65 fall annually [1] It is widely accepted that falls are multi-factorial in nature [2, 3] and that impaired balance function is associated with falls [4].

Page 6 - Line 15: Gait control or walking ability? Ambulatory control seems not appropriate.

This has now been changed to gait control.

Page 6 - Line 17: Recommend revising the terminology of inner ear balance dysfunction/exercise (all over the manuscript), since inner ear is an anatomical site, and the vestibular system is much more complex. It includes peripheral and central structures and functions. Particularly the compensation of vestibular dysfunctions is primarily dependent upon central processes.

The reference to inner ear has been removed, and has been replaced with vestibular system.

Page 6 - Line 18 - 19: The value of 80% might be inflated. Although the article by Liston et al., 2014 is a well-designed pilot study, the sample size is very small.

Similar studies (e.g. Jacobson et al, Agrawal et al) have demonstrated similar levels of impaired vestibular function, or utilisation of vestibular cues in older adults. These refs are already reported in the text (refs 4-7).

Page 6 - Line 48: Balance Physiotherapy is not a commonly used term (?). Consider revising to physiotherapy intervention targeting balance dysfunction or structured balance exercise program.

This has been changed to balance rehabilitation throughout.

Page 6 - Line 51: Overall, balance training may be easily performed if the applied protocol is simple, such as the Otago program. The prescription and execution of progressive balance exercises (intensity) is challenging, and include a combination of multisensory inputs, neuromuscular and coordination exercises, anticipatory postural adjustments exercises, and dual task tasks. Balance reactive strategies can be trained only in face-to-face interventions. Commonly, no all balance dimensions are accomplished in balance programs to prevent falls. Ultimately, I would recommend the authors to be cautious with the statement that balance exercises can be easily delivered or performed.

The line has been altered to: "Although exercises are typically brief and easy to perform". We agree that many of the facets of balance retraining are not currently implemented in falls rehabilitation programmes. This lack of comprehensive training has fed into the design of this intervention (which uses a multi-sensory training approach). We accept that reactive balance strategy training is most effectively provided in face-to-face settings using perturbations, but clinical protocols that use directed stepping are often implemented to retrain reactive balance. Numerous studies (and clinical interventions) have safely implemented multi-sensory interventions in older adults in the home environment (whether fallers or with vestibular impairment) e.g. Yardley et al., 2012, Liston et al., 2014, Baldursdottir et al., 2020.

Page 7 - Lines 3 - 4: Vestibular rehabilitation exercises are targeted to patients with vestibular problems. But, multisensory training that includes exercises that challenge the vestibular system by stimulating the integration of sensory information (re weighting of sensory inputs visual, vestibular and somatosensory) are usually included in balance programs during balance, gait and functional tasks. The problem is that these exercises programs are quite poorly described in the RCTs. Please clarify

what is the rationale for including specifically exercises targeting the vestibular rehabilitation for older fallers or clarify if these exercises would be in fact multisensory training.

This section has been modified slightly to read

“Similarly, rehabilitation programmes which address vestibular dysfunction (i.e. multi-sensory rehabilitation programmes) have been developed for older adults at risk for falls and have shown substantial additional reduction of falls risk to standard programmes.”

In addition, the description of the Active Intervention (HOLOBalance) (p10, line 20) describes that HOLOBalance is a Multi-sensory rehabilitation approach.

Page 7 - Line 19: Clarify what are the health care needs.

Otago program is a well-recognized and effective intervention to prevent falls, and for this reason was chosen as a comparator (standard care) in his study. The program is based on five home visits of a physio or nurse and requires a relatively small investment to be implemented. However, overall around 37% of participants remained exercising 3 times per week and 56% 2 times per week, below that what is recommended (80%). There are many barriers for attending group-sessions, and the supplementation of group sessions with additional home-based exercise is a strategy used in several effective trials for falls prevention. Alongside, the many technologies involved in the Holobalance platform it seems that one additional advantage would be keeping a long-term adherence.

We apologise for the lack of clarity. This section now reads

“The HOLOBalance tele-rehabilitation platform has been developed to address this lack of expert physiotherapists. HOLOBalance will provide a customised and interactive falls rehabilitation programme that incorporates: 1) functional balance training, 2) multi-sensory exercises to improve balance function and 3) cognitive-motor training, for older adults at risk for falls.

Page 8 - Lines 45 - 46: Please clarify if the 2 situations must be simultaneously present (FGA less than 22/30) and have fear of falling. Provide reference for the cut-off point as being one that reflects risk of falling.

This section has now been changed to read

-“ At risk of falls (i.e. FGA less than 22/30), and / or have significant fear of falling (FESI short form >10) and /or have experienced a fall/s in the last 12 months.”

The cut point for the FESI- indicates high concern about falling. This cut point is commonly used in clinical practice in the UK for people to enrol in falls rehabilitation programmes. This cut point is referenced in the subjective questionnaire section (Delbaere et al, 2010).

Page 8 - Line 51 - Please be more specific by what is meant by no significant visual, musculoskeletal or neurological dysfunctions. What specific conditions and or diseases will be excluded? (Parkinson disease, stroke, neuropathic diabetes)

The inclusion / exclusion criteria have been updated to improve clarity

Page 8 - Line 53: Any other acute problem/decompensation (such as acute vertigo, atrial fibrillation, congestive cardiac failure, etc) preventing exercises prescription. As this study is

a mutisensory intervention, it would be important to state if this pilot study will include only participants with chronic dizziness (and provide a definition).

This study will recruit all eligible older adults at risk for falls according to the inclusion / exclusion criteria. We will not specifically target older adults with chronic dizziness.

Older adults with dizziness of cardiac origin will be excluded, but older adults with stable cardiac conditions will be eligible to be enrolled.

Page 9 - Line 3: Depression is very common among people at high risk of falling. I agree that the engagement in a digital platform needs drive and motivation, but this is also a limitation that should be further discussed.

We acknowledge that this is a limitation for this study, and may exclude otherwise appropriate individuals for this research study. This criteria has been subject to peer review and approved by the funding body (EU).

Page 10 - Line 17: Please provide some information regarding entire duration of the session and particularly how much time will be spent in exercises in the orthostatic position; rest periods; and what kind of safety instructions the participants will receive (for example during the exercise that demands staying in a foam with eyes closed). Authors reported in the introduction that dual task activities would be accomplished in the HOLOBalance program. Please provide at least one or two examples of dual task exercises included. What kind of auditory tasks will be included (rhythmic)?

Please provide information regarding how participants will be familiarized with the technologies. It's expected that many of them will not have experienced exergames devices or augmented virtual reality devices.

Due to the limitations of the wordcount it is not possible to provide this level of detail within this protocol paper. We have provided a link to the study website which has further detail on the intervention.

Page 11 - Line 29: Clarify which questions will be addressed in the exit interviews and if some narrative analysis will be conducted with them.

This section has been clarified and now reads:

Exit interviews will be completed with participants that receive the HOLOBalance intervention to explore their experience of using the system, including perceived benefits, frustrations, and recommendations. Interviews will be recorded, transcribed verbatim and subject to thematic analysis.

Page 11 - Line 60: All secondary outcomes will be used to make between group comparisons? This is a proof of concept study and the effectiveness of HOLOBalance will not be measured. If, a future clinical trial will be conducted would be important to define clinical important differences or meaningful differences for important outcomes rather than effect size.

Clinical variables, such as the number of medications, use of psychotropic and medications, numbers of diseases, number of falls in the previous 12 months is going to be reported?

The proposed statistical approach has been reviewed and approved by our funders. We will consider these points in our future analysis.

Page 15 - Lines 59 - 60: Which measures and cut-off points are going to be used to stratify

subgroups.

Please clarify what would be the content of the validation report.

We apologise for the confusion this section caused. It has now been removed from the manuscript.

Authors did not provide a discussion section with the possible impacts and clinical implications of their study.

The authors followed the BMJ guidance for reporting study protocols. These do not require a general discussion section for protocol papers.

Table 2 would be better displayed in a study flowchart.

the format of table 2 has been altered to improve clarity.

Reviewer: 2

Reviewer Name: Eric Anson

Institution and Country: University of Rochester, NY USA

Please state any competing interests or state 'None declared': None declared

Please leave your comments for the authors below

The manuscript "Investigating the Feasibility and Acceptability of the HOLOBalance System Compared to Standard Care in Older Adults at Risk for Falls: Study protocol for an assessor blinded pilot randomised-controlled study" is a well written description of a proposed clinical trial investigating the feasibility and acceptability of HOLOBalance for training balance in older adults within their home. This is an important study addressing a need within the aging community. There are a few minor points that will benefit from further clarification which are detailed below:

Page 5 lines 7-11 the 2 phrases are poorly connected, please re-write to improve clarity.

This section has been redrafted to improve clarity and now reads.

"This is despite promising evidence suggesting that combined cognitive and functional training may provide improvements beyond single task training [38-40]. Similarly, rehabilitation programmes which address vestibular dysfunction (i.e. multi-sensory rehabilitation programmes) have been developed for older adults at risk for falls and have shown substantial additional reduction of falls risk to standard programmes [41-43]. These programmes require customised, expert led interventions to optimise recovery, however there is limited availability of experts to provide these individualised interventions and these may hinder their translation into clinical practice."

Page 5 lines 20-26, if possible please re-write into 2 sentences. The length of this statement as written makes it hard to follow.

This section has been modified and now reads

"The HOLOBalance tele-rehabilitation platform has been developed to address this lack of expert physiotherapists. HOLOBalance will provide a customised and interactive falls rehabilitation programme that incorporates: 1) functional balance training, 2) multi-sensory exercises to improve balance function and 3) cognitive-motor training, for older adults at risk for falls."

Methods page 6, are the authors going to screen prospective participants with the FGA, or is the FGA routinely administered in the settings from which subjects will be recruited?

The inclusion / exclusion criteria for this study have been modified to improve clarity. For inclusion into this study participants will be screened using the FES-I (SF), falls history and FGA.

Methods page 7 line 10-11 - please clarify and consider the potential for sample bias that this specific inclusion criteria creates

The inclusion / exclusion criteria for this study have been modified to improve clarity. The requirement for space is necessary to provide a safe environment for using the system.

Methods page 7 lines 42-45, this statement is unclear

This has been clarified

Methods page 8 lines 3-13, please clarify how many outcome assessors will be used at each site, and what if any training each assessor will have. Also please clarify what steps have been taken to ensure similar grading for instruments that are open to interpretation because of subjectivity.

This section has been modified to increase clarity and now reads:

“No out of hours randomisation is required for this study.”

Page 9, lines 49-50 please clarify how and when adverse events will be identified.

This has been clarified in the text and now reads:

“Safety of the HOLOBalance intervention will be assessed by recording and assessing all adverse / serious adverse events and device effects (and reporting these to appropriate agencies as required). These will be reported by participants to their treating clinician or by self-reporting on the HOLOBalance database.”

Methods Page 10, please provide information on psychometrics of the mini-BEST and FGA relevant to the study population if available.

This has been added to the text.

Methods Page 10, please clarify the definition of a fall.

This has been added to the text and now reads:

“Falls diaries will be used to record whether participants have fallen each day, with falls defined as a person coming to rest inadvertently on the ground or floor or other lower level. Participants will return falls diaries monthly for the duration of the intervention and for 6 months after completion.”

Methods Page 10, please clarify that validated version of the MoCA exist in the languages appropriate for the proposed study populations. Same comment for CANTAB, and all subsequently listed questionnaire based outcome measures.

The MoCA and CANTAB are both validated for use in Greek and German languages. Where possible, validated translations of measures will be used, and where not possible translated versions will be used.

Methods Page 10, what is "MCI169"

This typo has been deleted

Methods page 13, perhaps "entered" instead of "inputted"

This has been changed as per the reviewers suggestion.

Methods page 13 lines 52-57, perhaps consider consolidating this statement to something like "Sex/gender will be considered as a covariate biological variable for all analyses.

This section has been removed

There is a discrepancy in the proposed sample size on page 6 (120) and page 13 (80) and page 14 (120), please clarify.

We apologise for this mistake. This has been rectified.

Page 14, PPI and REC are used without being defined.

These terms have now been clarified in the text.

Page 14, please clarify the role of the funding agency/industry partner with respect to data dissemination. Specifically clarify whether the "Holobalance data management board" has any oversight regarding approval of data/results dissemination.

This section has been altered and now reads

"Electronic data will not be made publicly available and access will to the dataset will only be provided by the data management board of the HOLOBalance research consortium. Use and re-use of the pilot dataset will be subject to the license under which the data objects were deposited."

The board are made up of senior members of the research team. They do not have oversight in the release of data / results, but act as the gate keepers to the dataset to ensure that data is not released into the public domain until all relevant works have been completed.

Table 1: Seated gaze stabilization exercises, please clarify the target distance for pitch and yaw.

Please clarify if the "Step turn through

180°" is a voluntary or cued turn. Table 1, this progression is unclear "Increase head turn amplitude when visual targets are removed," how will the subject know how far to turn if the targets disappear.

- The target distance for gaze stability exercises can be varied in the computer programme.
- The 180 degree turn is a voluntary turn in response to a verbal command to initiate the turn
- the text in the table 1 has been altered for clarity and now reads "Turn head fully to side as if crossing the street."

Reviewer: 3

Reviewer Name: Yoshiro Okubo

Institution and Country:

Neuroscience Research Australia
Australia

Please state any competing interests or state 'None declared': None declared

Please leave your comments for the authors below

This manuscript describes the study protocol for the pilot feasibility trial using the augmented reality program called HOLOBalance involving remote coaching of strength and balance training. The intervention has great novelty, the manuscript has been written very well and worth publishing in this journal. I have a few minor comments.

I believe that reproducibility is an important aspect of a scientific article. Information on the development or reference to it would be beneficial. If the authors do not wish to publish such information due to intellectual property, their plan on how to make the final product accessible can be mentioned.

The development of the HOLOBalance system is a research in action project, and as such the developmental work is currently being written up for publication. HOLOBalance will be commercialised in the future if it is shown to be effective in future definitive studies.

Please include a reference for the OTAGO HEP booklet and the frequency of home visits to the control group participants.

The reference has been added, and frequency of home visits has been added to the text.

Readability of Figure 2 should be improved.

We have generated a new version of the figure.

The trial was due to start in April 2020, please provide an update on this.

We anticipate that the trial will start recruiting in mid-end September 2020 (dependent on COVID-19). The current delays are due to the research institutes being closed due to COVID-19.

Please provide figure legend/caption

These have now been inserted into the document.

VERSION 2 – REVIEW

REVIEWER	Eric Anson University of Rochester, USA
REVIEW RETURNED	31-Aug-2020

GENERAL COMMENTS	All comments have been adequately addressed in this revision.
---

REVIEWER	Yoshiro Okubo Neuroscience Research Australia, Australia
REVIEW RETURNED	16-Sep-2020

GENERAL COMMENTS	I have no further comments.
-----------------------------